# Male Victims of Sexual Assault: A Review of the Literature

**DOI:** 10.3390/bs13040304

**Published:** 2023-04-03

**Authors:** John C. Thomas, Jonathan Kopel

**Affiliations:** 1Department of Counselor Education & Family Studies, Liberty University, Lynchburg, VA 24515, USA; 2School of Medicine, Texas Tech University Health Sciences Center, Lubbock, TX 79430, USA

**Keywords:** male rape, sexual assault, sexual trauma, rape myths, counseling

## Abstract

Compared to female rape victims, the literature addressing male rape victims remains a growing area of interest for counselors and scholars. This article aims to review the growing literature on male sexual assault victims. Specifically, the review will examine the literature on male victims of sexual assault in nine sections: (a) an overview of male sexual assault, (b) male rape myths, (c) prevalence, (d) responses to male victimization, (e) populations and perpetrators of male victimization, (f) risk factors, (g) reporting, (h) the impact of sexual assault on men, (i) help-seeking, and (j) implications for counseling. Empirical studies, case reports, and books are included in the review.

## 1. Introduction

Sexual assault, harassment, non-penetrative acts of sex, and attempted and committed rape are all examples of violent sexual behavior. Sexual victimization, according to the Centers for Disease Control and Prevention, is a sexual act that is conducted or attempted by another person without the victim’s freely given consent or against someone who is unable to consent or refuse. It is a significant public health issue that warrants significant public, societal, and judicial attention [1]. According to current estimates, over 27% of men and over 32% of women had been sexually victimized at some time in their lives [2]. Molestation is abusive sexual activity committed by one person against another [3]. It is frequently committed with the use of force or by taking advantage of someone [3]. Molestation is a term used to describe a single incident of sexual assault on a young child, whereas sexual abuse is a term used to describe a pattern of repeated sexual assaults. Sexual abuse can have catastrophic and pervasive repercussions.

Victimization has a direct negative influence on the physical and mental health of those who experience it, leading to short- and long-term physical injury, fear, anxiety, despair, post-traumatic stress disorder (PTSD), low self-esteem, social difficulties, and suicide ideation [4]. Together with other socioeconomic implications such as being unable to work, dropping out of school, being stigmatized, and being shunned by their communities, it is also linked to an increased risk of sexual and reproductive health issues [5,6,7]. A sexual act that is conducted or attempted against a person who is unable to consent or refuse by another without the victim’s free consent is known as sexual victimization. Men are generally seen to be as less affected by sexual victimization. Yet, there is evidence to support the idea that sexual victimization affects male victims mentally just as much as it does female victims and may even be linked to worse results [8].

In ancient and modern societies, female sexual assault has remained a prevalent social and public health challenge [9]. However, the infrastructure and support to allow reporting of sexual assault was extremely difficult [10]. Specifically, “Few women were willing to endure the damage to reputation and prospects for marriage that followed from bringing a rape complaint, and if they did, the prospects for vindication of their complaint were remote indeed. The common law required a woman claiming rape to make a highly scripted showing that sexual relations were nonconsensual; she had to show that sex was coerced by force and against her will -that she succumbed to overpowering physical force despite exerting the ‘utmost resistance’” [11]. Despite the social stigma against sexual violence perpetrators, female sexual assault was permitted in several cultures depending on the victim’s social standing or violation of social taboos. For example, the sexual coercion of enslaved African American women was permitted without any legal repercussions [12]. Furthermore, clinical investigations into the prevalence, psychological trauma, and therapy for female sexual assault victims remained unknown until the 19th and 20th centuries [13].

Although the female victims of sexual assaults have garnered a great deal of necessary attention, the body of literature examining men as victims of sexual violence is lacking [14,15,16,17,18]. Consequently, what is known about adult male sexual victimization (AMSV) is dwarfed by the knowledgebase on female victimization [19]. It is estimated that the help and support for male victims is over 20 years behind that of female victims [20]. Furthermore, male victims have fewer resources and greater stigma with female sexual assault victims [21]. Approximately 20–25% of female sexual assault incidents are reportedly reported in the United Kingdom (UK) [22]. For instances of AMSV, it is anticipated that this statistic is far lower. Prior to 1994, the United Kingdom’s legal definition of rape was restricted to instances of forceful or unconsented vaginal penetration, thus excluding male victims [23]. Instances of forced or unconsented anal penetration fell under the legal definition of “buggery”, which was punishable by a significantly lighter fine. Between 10–20% of female sexual assault victims in the United States (US) are believed to have reported the crime, and the number of male victims is likely to be far lower. All but three jurisdictions in the United States now have gender-neutral rape laws, with Georgia, Mississippi, and Idaho being the three exceptions.

During the 1980s, there has been a significant increase in the amount of literature on the topic, indicating that AMSV is not as uncommon as the former paucity of material on the issue suggested [4,24]. The incidence and prevalence of sexual assault against adult men, the reasons behind it, and the psychology underlying victims’ involuntary sexual reactions are all covered in this article’s examination of the literature. Due in part to the misconception that the victim’s erection or ejaculation during the assault constitutes permission, the legal system has been reluctant to offer male sexual assault victims a legal redress, despite the increased awareness of these crimes [4,24].

Although the literature on sexual violence against women is laudable and although it has provided a foundation upon which to explore male victimization [25,26], the latter is an experience that is worthy of investigation. Even though adult male sexual violence (AMSVo) is becoming more widely acknowledged as an issue, the literature unanimously agrees that there is a dearth of information on the subject when compared to information on female victims [27].

This article provides a review of the literature on AMSV. First, we provide the background for our literature search and criteria. Next, a discussion of the numerous definitions is offered on AMSVo and related terms. We examine the literature related to prevalence and barriers for men to report incidents of sexual violence. Biases and misconceptions which impact both the reporting of an incident and response to men who are victims of sexual violence are explored in depth. Research is provided on typical male responses to sexual victimization, populations of men most at risk and risk factors to being violated, the emotional impact on men, help-seeking by men, and implications for treatment to equip counselors with required knowledge on AMSV and to empower them to address challenges facing male victims. Finally, the limitations of this review and recommendations for future directions in this research field are outlined.

As will be evident from this article, much of the literature is old, dating back to the 1970s when seminal research was taking place. Since that time, empirical investigations of male sexual victimization have only been sparsely carried out over the course of nearly 50 years. Given the lack of research into adult male sexual victimization, we do not have certainty regarding what is applicable to today’s victims and what is outdated. According to this review, men frequently have erections or ejaculate uncontrollably during a sexual assault; nonetheless, the victim’s response does not imply consent. This review also supports the notion that anal rape might cause men to experience involuntary erections or ejaculations.

## 2. Definition of AMSVo

Several terms describe sexual violence, including *sexual victimization*, sexual assault and rape [28,29], although the literature often uses the terms interchangeably. Adult sexual assault refers to all types of sexual assault, including rape, which is defined as the penetration of the victim’s mouth or anus by a penile, digital, or foreign object while using force, coercion, violence, threats of violence, or incapacitation. It covers oral-genital groping, fondling, kissing, groping, and any other type of unwelcome sexual contact carried out in similar ways [30]. The definition specifically mentions actions of unwanted female-initiated sexual contact because there are still misconceptions about the subject. While determining the issue of prevalence, it is crucial to be aware of the full range of sexual assaults, even if it is acknowledged that non-rape forms of sexual assault could not have the same effect on the victim as an attempted or successful rape. One in thirty-three men have reportedly been the victim of an attempted or successful rape, but when the definition is expanded to include sexual assault, the figure rises to one in five men [30,31].

Currently, there have been several definitions used to define sexual assault. Isely and Gehrenbeck-Shim (1997) defined male sexual assault “as any non-consensual sexual acts perpetrated against a man, 16 years or older, by a male or female” (p. 160) [32]. In 2020, the Department of Justice’s (DOJ) definition of sexual assault included a range of victimizations and was separate from rape or attempted rape. It includes attacks or threatened attacks involving unwanted sexual contact between victim and offender, with or without force; grabbing or fondling; and verbal threats. Additionally, they define rape as the “penetration, no matter how slight, of the vagina or anus with any body part or object, or penetration by a sex organ of another person, without the victim’s consent” (https://www.justice.gov/archives/opa/blog/updated-definition-rape; accessed on 28 March 2023) [33]. The World Health (WHO, 2002) expands the definition beyond physical contact: “any sexual act, attempt to obtain a sexual act, unwanted sexual comments or advances, or acts to traffic, or otherwise directed, against a person’s sexuality using coercion, by any person regardless of their relationship to the victim, in any setting, including but not limited to home or work” [27,34].

Before the 2003 act completely altered the legislation on sexual offenses, the rape of a man was covered by the 1994 amendment to the Sexual Offences Act 1956 [35]. Until 1994, there was no such thing as “rape of a male” in English law; instead, there existed a felony called “buggery” (unconsenting penile–anal penetration), which had a lesser punishment. The relatively recent recognition of rape of a male being possible and illegal is reflected in how this issue is denied or misunderstood within society [35].

Although no definition is specific to men, we are satisfied that the DOJ’s definition of sexual assault and rape provides the clearest distinction between sexual acts. The WHO’s definition includes many acts not addressed in the DOJ’s definition, such as attempts to obtain sexual acts, verbal comments, and advances. These are not considered sexual violence as specified in the DOJ’s definition but are noteworthy for understanding the experiences of men victimized by them. The WHO includes behaviors that would be examined in sexual harassment and sex trafficking literature. Of importance, however, is that the WHO captures the reality that all the identified behaviors can occur regardless of the relationship between offender and victim. For research purposes, clearer operationalized definitions are needed to focus on the unique aspects of each type of behavior and analyze distinctions between them.

## 3. Prevalence of AMSVo

In Western nations, such as the UK, the US, and the Nordic countries, the prevalence of male-on-male rape or sexual assault is believed to be between 5 and 10% of all sexual assaults each year [35,36,37,38]. According to the Home Office crime statistics for England and Wales, there were 9901 rapes of victims aged 16 and older in total during the 2010–2011 financial year, 9509 of whom were female (96%) and 392 of whom were male (4%). This demonstrates the relative rarity of the problem and suggests why forensic doctors, therapists, and other treatment providers would not see many male patients. The ratio of male to female patients at the Saint Mary’s Sexual Assault Referral Centre in Manchester, UK, illustrates that although this category is smaller than that of female victims, it is one that has been expanding. The Centre saw fewer than 20 men who had been the victims of an acute sexual assault (i.e., not past, or ongoing child sexual abuse) in the first five years of its operation (1986–1991) [35,39]. By 2002, that number had increased to over 40 annually. Fewer than 30% of men from those first five years were referred by the police, whereas this number was over 70% in 2002. This surge in referrals was mirrored by a growth in referral types [35,39]. Similar comparative hesitation was observed in Norway, Denmark, Iceland, and Finland among men who reported anything at all, particularly to the police. These statistics are a result of the frequency and sorts of assaults themselves, as well as the various responses that men have when discussing what has happened to them [35,39]. 

Many studies have noted the difficulty of obtaining reliable and accurate statistics on adult male victimization [40,41]. Several factors, including definitional limitations, may cause different studies to have different prevalence findings. These include sampling methods, how objects are written on scales, and the aforementioned “unrecognized assault” [8]. Elliot and colleagues (2004) examined prevalence statistics for adult men as a whole and found that it ranged from 0.6 to 8.3% [42]. This range of data is startlingly wide and might not accurately reflect the population’s overall victimization rate. Although it is acknowledged that some adult male subpopulations are more vulnerable to victimization, none of these subpopulations—with the probable exception of those in correctional settings—appears to have received enough attention in the research to paint a full picture. However, even though there is a greater amount of research on male prisoners, the most of it seems to be older, indicating that any attempt at synthesis would probably result in outdated and unhelpful data [30]. Men attending college appear to be the second-largest research group. Although the literature may be scant, it does seem to show indications of a prevalence issue significant enough to motivate more study for this demographic. According to some estimates, college-going men between the ages of 18 and 24 are five times more likely to be assaulted than their non-college counterparts, placing them at the greatest risk of victimization [30].

Furthermore, the prevalence reports of male sexual assault vary drastically depending upon the study. Stemple and Meyer (2014) found high prevalence rates of male victimization, approaching that of women, after reviewing five independent surveys by two federal governments [43]. The national crime statistics show 10% of rape victims or 1 in 33 men (3%) have experienced rape [28]. Although the rates of those reporting unwanted sexual contact or pressured intercourse have been reported in the ranges of 38 to 48% for male college students [44], incidence approximates 4% in most studies [42]. Based on the national Criminal Victimization 2019 survey, the percentage of violent victimizations reported to police was higher for female victims (46%) than for male victims (36%). This difference can largely be attributed to reporting of simple assaults, as the percentages of violent victimizations reported that excluded simple assaults were similar for women (47%) and men (46%) [45]. Additionally, male victims reporting unwanted sexual contact or pressured intercourse has been reported in the ranges of 36% to 46% [45]. Among male prison inmates, 59% of male inmates reported some form of childhood sexual abuse. It is likely, however, that the documented rate is likely an underestimation of the magnitude of the problem. Studies also show mixed results regarding who is most at risk. For example, Coxell et al. (1999) reported a higher prevalence in the homosexual male community [46], whereas Isely and Gehrenbeck-Shim (1997) found that heterosexual men are more likely to be victimized (71.4%) [32]. Further, Isely and Gehrenbeck-Shim found that most victims were young (ages 16 to 30) and Caucasian (85%). Whether heterosexual or homosexual, the literature suggests that any man can be a victim of rape [13].

Recently, other studies have estimated the prevalence of sexual assault among men at college campuses [47,48]. In a study comparing students from sexual minorities and non-sexual minorities, Edward et al. estimated the six-month incident rate of three types of intimate partner victimization. This was a large-scale study (N = 6030) encompassing eight New England universities, but only 34.1% (N = 2055) of the sample were male college students. The respondents were distributed equally across years in school [47]. According to their research, 7.4% of the male students (15.5% of sexual minority students and 6.5% of heterosexual students) reported having been the victim of sexual victimization at least once in the previous academic year.

In addition, Ford et al. aimed to address the dearth of information on sexual victimization for groups other than heterosexual female victims [48]. The Online College Social Life Survey dataset was used in this study to evaluate the victimization rates of undergraduate college students (N = 21,000) from 21 four-year universities, 6581 of whom were men. Three dichotomous questions that focused particularly on sexual encounters that had occurred since they started college were used to measure sexual assault [48]. The authors enquired as to whether someone had ever physically forced them into having sex, attempted to physically compel them into having sex, or engaged in sex with them against their will while they were physically incapacitated. According to their findings, roughly 13% of straight college men, 24% of homosexual undergraduate men, and 17% of bisexual undergraduate men reported having experienced at least one of the three types of sexual assault classified in this study [48]. The most frequent kind of attempted sexual assault among straight and bisexual men was around 6% and 17%, respectively, but homosexual men reported more instances of having sex while unconscious (around 14%) [48].

## 4. Barriers to Reporting AMSVo

Male and female victims may decide not to disclose information to protect a friend or family member, out of concern of retaliation by their attacker, and out of concern that they will be blamed personally for their victimization [42,49]. Men may also choose not to report if they have issues with their sexuality. Men are more likely to encounter reporting issues relating to their sexuality given that they are more likely to be victims of other men. Men who have not publicly acknowledged that they are anything other than heterosexual may choose not to report for fear of having to come out. In addition, heterosexual men who were assaulted by other men might not come forward for fear of having their sexual orientation revealed. Another specific rape distortion that affects men is the belief held by the public and healthcare professionals that men cannot be raped [35]. Men may opt not to complain if they are simply going to be informed that what occurred to them did not occur—an invalidation of their experience. A lack of knowledge regarding the physiologic reaction to attack and the fact that erection or orgasm can occur even in traumatic situations may contribute to this belief. This false notion may not only prevent people from believing that men may be abused, but it may also prevent men from recognizing victimization when it does.

The literature strongly suggests that both adult men and women underreport sexual violence to law enforcement and medical services, and research consistently conveys that men are less likely to report [50,51,52,53,54]. Approximately 90 to 95% of all male sexual violations are not reported [55]. Walker and associates reported that 12.5% never disclosed their assault to anyone; among those who did, 54% delayed reporting for at least one year [56,57]. In their study, four of the five men who reported their assault to the police regretted their decision. Victims said that not only were the police unsympathetic and disinterested, but even more traumatic than the actual victimization. In fact, one victim described the legal process as having “had a worse effect on him than the rape itself” (p. 75).

Underlying the failure to report are issues of stigma, shame, guilt, embarrassment, fear of ridicule or not being believed, concern over confidentiality, and concern of having their sexual orientation questioned [49,58]. Men tend to believe that authorities will not believe them or will make light of their victimization. Graham (2006) and West (2000) suggest that fear of homophobic reactions and a dissolution of their masculinity deters men from reporting. Walfield (2018) argued that men fear being marginalized. Furthermore, Page (2008) notes [59,60,61]:

“If police officers endorse stereotypical ideas about gender and rape only cases meeting the characteristics of an “ideal” rape (i.e., the victim and offender are strangers, the victim incurs physical injuries and there is a physical evidence of sexual assault) will be deemed credible and will thus be investigated...the more a victim, or the characteristics of an assault, deviate from this preconceived idea, the less likely police and prosecutors are to devote extensive time and energy to processing the case” (p. 45).

Victims are also hindered by whether the act fits their own a priori definition of sexual victimization. Warfield (2018) believes that men may use their perception of the act as a defense mechanism from acknowledging their own victimization in the first place [59]. It mitigates feelings of vulnerability and fear, whereas offenders use sexual stereotypes to justify acts of sexual violence to minimize their own actions and the harm done [59]. Burt and DeMello (2005) captured the bottom line of misconceptions when they asserted that these are prejudicial, stereotypical, and false beliefs about rape, rape victims, and rapists. Another reason for the lack of reporting is offered by Campbell (2008), who suggests that the low conviction rates contribute to reluctance [62]. Mixed results have been found regarding the extent of the violence as a factor in reporting. Frazier (1993) found that reporting to medical services typically required extreme circumstances such as gang rape [63]. In contrast, Monk-Turner and Light (2010) found that men who experienced penetration were markedly reluctant and less inclined to report the assault [53].

Developing knowledge and understanding through empirical research to inform support providers so that they are aware of what contributes to the blame attributions of male victims is a necessity. The underreporting of sexual victimization by men has served to limit the ability to competently research the topic. James (2018) noted that self-incrimination, trepidation, and shame hinder adult male victims from disclosing their experience to researchers [64]. Thus, the information available is from those who choose to disclose their experience or from governmental organizations and other agencies that have a connection to victims.

Although research affirms that male sexual violence is a legitimate problem [24,40,60], its reality remains virtually invisible to the public [40]. Researchers found numerous “myths” or misconceptions/biases that obfuscated male victims from being accepted and understood [40]. The literature suggests there are several misconceptions associated with AMSV. Kassing and colleagues (2005) noted four misconceptions: (a) it is rare, (b) women cannot be perpetrators, (c) only happens in prison, and (d) men do not suffer psychological consequences [65]. Additionally, Struckman-Johnson (1991) identified one additional misconception [66]: men are too strong to be forced into unwanted sex. Pino and Meier (1999) also noted the erroneous belief that male victims lose their manhood [67]. Stemple and Meyer (2014) examined public misconceptions on male rape and identified three factors: traditional gender stereotypes, outdated conceptions of rape, and methodological sampling biases that exclude inmates [43]. The Center for Disease Control and Prevention (CDC) and the Federal Bureau of Investigation (FBI) found that men and women reported a similar prevalence of nonconsensual sex (women: 1.270 million; men: 1.267 million).

Furthermore, the idea of the “ideal victim” is still widely used within this gender-based assumption of sexual victimization research [4]. In this conception, the perfect victim of sexual assault is conceived to be one who society is most likely to label as a victim. Within this framework, a person must fulfill five requirements in order to qualify for this status: (1) be weak; (2) be engaged in a respectable activity; (3) be somewhere she could not possibly be “blamed” for being; (4) the offender must be big and bad; and (5) the offender must be unknown to the victim and have no personal relationship with her. The emphasis in this depiction is on men as sexual offenders and women as victims. In addition, some individuals think that rape is an act of dominance and control towards women, which prevents men from declaring themselves to be sexual victims. Nonetheless, there is a serious issue regarding male sexual victimization, which takes place in a variety of venues, such as homes, workplaces, schools, on the streets, in the military and in times of conflict, as well as in jails and police custody [4].

### 4.1. Blaming the AMS Victim

Male victims are perceived to carry a level of blame for not resisting their attacker. Others may question how a man can achieve and maintain an erection and sexually perform if the sexual encounter is a coercive situation. Smith and colleagues (1988) a assess 77 men and 89 women who made a series of judgments about two randomly cases as if they were on a jury [68]. A MANOVA analyzed a 2 (sex of victim) × 2 (sex of assailants) × (sex of subjects). The attributions of victim responsibility found that men were more likely to be assigned blame than female victims. They also found that when the assailant was a female less impact was reported than a male-to-male act. It is hypothesized that if men become aroused and sexually respond to the perpetrator, they wanted and enjoyed it. Given the strong feminist theoretical paradigm and feminist ideology [69], women are classified as victims and men as offenders. Javaid et al. highlight this point by noting some feminists reject male rape to validate women’s experience of sexual violence by viewing men as solely offenders [51]. The perceived minimal force needed to overpower female victims mitigated the perceived impact on men. Male victims note that the implication is that “the use of force determines concern about victimization” (p. e20).

Male victim bias perpetuates victim blaming. Specifically, the acceptance of male rape was a strong predictor of victim blaming, suggesting that acceptance of stereotypical ideas about male rape means that a person is more likely to engage in male victim blaming behaviors. This is of concern, particularly as it is likely that these falsehoods regarding male rape are accepted widely [70], as confirmed within this study with high levels of acceptance of certain misconceptions or false beliefs on male rape. This finding is supported by Kassing et al. (2005) and Johnson et al. (2006) [65,66].

### 4.2. Accusations of Homosexuality against Adult Men

Another masculine-related misconception is that men are sexually assaulted by homosexuals, perpetuating the false belief that sexual violence is about sex and only committed by homosexual men and victims are primarily homosexual [51,71,72]. Similarly, it is often thought that women do not assault men, leaving the perpetration of male sexual victimization to men. Accepted stereotypes that homosexuals solely assault men and only gay men are victims facilitate the inaccurate understanding and poor handling of male victims [51,71,72].

### 4.3. Female Perpetrators of AMSVo

The first systematic report on adult male victims was by Sarrel and Masters (1982) who interviewed men who reported women perpetrating sexual assault [24]. The researchers noted psychological distress, post-trauma reactions, and impaired sexual functioning. Later studies found that gay men are more likely than women to have pro-victim judgements and endorse male rape falsehoods, including victim blaming [56,57,66].

### 4.4. Toxic Masculinity, Macho Imagery, and Their Consequences on Men’s Health and Life Habits

Stereotypical ideas about men calls into question a man’s masculinity when sexually victimized [40]. The incongruity between historic beliefs of masculinity linked to strength and AMSV maintains the bias of invulnerability. Not only do adult male victims believe that they will be viewed as gay, but they also question it themselves [40,58,73]. For heterosexual men, being assaulted by another male is likely their first same-sex encounter. The horror of doubting their own sexuality leads to self-loathing and hatred of homosexuals [40,51,71,72,74]. As a result, men are more likely to express anger than women. Several studies suggest that adolescent men who strongly adhere to masculine norms may be more likely to commit violence [75,76,77,78,79]. The social perception of men’s actions, emotions, thoughts, and behaviors, as well as their rights and responsibilities in society, is described by traditional masculine standards. According to several sociocultural and psychological theories, socially constructed masculinity may help to explain why men commit violence. Nonetheless, not all men who strongly uphold these archaic masculine ideals will resort to violence against female partners. These results imply the necessity for a sophisticated approach to comprehend the function of conventional masculine standards and men’s use of AMSVo [75,76,77,78,79].

One of the most common sociological theories to describe toxic masculinity or the macho personality constellation was through the script theory by Tomkins [75,76,77,78,79], which comprises three behavioral dispositions supported by the following beliefs: (1) entitlement to callous sex, (2) violence as manly, and (3) danger as exhilarating. The scene serves as the fundamental analytical unit in script theory. Specifically, the theory references an occurrence in a life that has already been lived, ordered by at least one affect and its object, and designated by a beginning and an end, and a series of interrelated events form the narrative of a life that we refer to as a personality. A script uses a set of guidelines for understanding, reacting to, defending, and producing comparable scenarios to connect and organize the information in a family of related scenes. According to the set of guidelines in the macho script, the macho man generates, interprets, and reacts to circumstances that threaten, challenge, or provide chances for him to play out his role as a macho man [75,76,77,78,79]. These scenes were arranged according to their effects and the rules for reading, comprehending, making, and justifying situations that they developed while being socialized into hypermasculine scripts. The effect is the main driver in people. Tomkins defines affect as a collection of facial-specific muscle and glandular responses that are widely dispersed throughout the body and produce sensory feedback that is either essentially “acceptable” or “unacceptable”. Innate and kept in subcortical areas are the programs for discrete affects. Affects are originally organized and activated by innate scripts, but learnt scripts are what create the dynamic complexity of human experience and motivation. The freedoms of affect in time, intensity, the density of investment, the choice of objects, and investment in possibilities, as opposed to drives, are what allow humans to worry about nearly anything with varying levels of urgency [75,76,77,78,79].

According to Tomkins, the psychological processes occurring within the scene itself, such as perception, cognition, and action, are amplified by affect. Yet, strong emotion may only have a fleeting impact, with little or no influence on personality. Scenes must be psychologically exaggerated to cause an urgent search for a system of rules to organize their emotionally compelling content [75,76,77,78,79]. When a narrative review of scenes occurs in the consciousness, psychological magnification is the further amplification of the set of already magnified scenes by fresh emotion. The rules for understanding, producing, anticipating, and manipulating the scenarios as well as the scenes themselves are magnified psychologically. For the macho man, it is the effect of interest–excitement that psychologically increases the affectively urgent search for guidelines on how to be a “true man”, experienced when he consciously evaluates memories and thoughts of possibly dramatic scenes [79]. If he is brave enough to take on its challenges, this dangerous world might even thrill him. Similar to language, personality can be seen from a synchronic or diachronic perspective. Personality can be seen as a posture or position that describes a group of potentialities that are currently unrealized. The macho is gradually conducting his life according to his macho script. The macho script must be lived out to honor the machismo philosophy [75,76,77,78,79].

The macho man is the nomadic warrior’s cultural offspring; the macho philosophy is a warrior’s philosophy. The macho warrior rules over whatever he has taken, serving as patriarch and ruler. His wives, children, and slaves are his property and owe him devotion and respect. The macho man must be willing to take significant risks to preserve that dominance [75,76,77,78,79]. He must also be willing to use violence to subdue rival men and callous sex to subjugate female foes. Being a male within a macho culture socially inherits the ideological script of the macho man. Affects are categorized in American society as either “superior and masculine” or “inferior and feminine”, which are antagonistic contrasts [75,76,77,78,79]. To establish the ideologically desired gender-stereotypic contrasts of masculine and feminine, the act of assigning to the male or female sex tends to bias the socialization of affects into gender scripts that separate and stratify men and women through this division of human emotions. Macho ideology denigrates those who demonstrate “inferior, feminine” affects while honoring “superior, masculine” affects. To achieve hostile–dominant interpersonal goals driven by the effects of excitement, wrath, disgust, and contempt, macho scripts thereby enhance masculine gender role behavior. The macho must be hypermasculine in thought and deed, not simply male and not just masculine [75,76,77,78,79].

The “superior masculine” gender script has in fact become a cultural ideology when it elevates both patriarchal supremacy as the ideal political and familial value and antagonistic physicality and toughness as the core of masculinity [75,76,77,78,79]. The most significant group of scripts is the ideological one because of its breadth, abstraction, and ability to give fact meaning and emotion. Ideologies are the main forces behind grouping, differentiation, and division since they stand in for the diverse religions by which people live and die. Men who adhere to the macho philosophy are bound together in male honor societies, the sexes are distinguished by male domination and female submission, and society is split into the strong and the weak based on how well each group of people has embodied the values of “true masculine superiority”. By assuming that physicality, virility, and masculinity are the ideal essence of real men who are adversarial warriors competing for scarce resources in a dangerous world, we define the ideology of machismo as: a system of ideas forming a world view that chauvinistically exalts male dominance [75,76,77,78,79]. Parents’ training of male children into a heightened, hypermasculine gender script known as the “macho script” is sanctioned and encouraged by the machismo cultural ideology. Ideological scripts are a type of script that simultaneously validate and fulfill themselves. As they lessen the temptation of relaxed enjoyment and communion and prevent the experiencing of the unmanly effects of fear, distress, and shame, these scripts interpret, predict, control, replicate, and evaluate the manly effects of surprise, excitement, anger, disgust, and contempt for the inferior foe. This is what it means to be a macho man [75,76,77,78,79]. In summary, the seven key elements of the macho man include:Unrelieved and unexpressed distress is intensified by the socializer until it is released as anger.Fear-expression and fear-avoidance are inhibited through parental dominance and contempt until habituation partially reduces them and activates excitement.Shame over residual distress and fear reverses polarity through counteraction into exciting manly pride over aggression and daring.Pride over aggressive and daring counteraction instigates disgust and contempt for shameful inferiors.Successful reversal of interpersonal control through angry and daring dominance activates excitement.Surprise becomes an interpersonal strategy to achieve dominance by evoking fear and uncertainty in others.Excitement becomes differentially magnified as a more acceptable effect than relaxed enjoyment, which becomes acceptable only during victory celebrations.

These elements of the macho men mentality/ideology can lead to a version of toxic masculinity [75,76,77,78,79]. Most studies characterize toxic masculinity, in part, as a collection of behaviors and attitudes that may involve hiding or suppressing suffering, maintaining a façade of toughness, and possibly even acting as a sign of dominance. The truth is that men have been pressured to be particularly tough due to the pervasive fear of being labeled as “feminine” or “weak” in many cultures. In this respect, toxic masculinity requires men to not display their emotions publicly or openly. We can therefore conclude that toxic masculinity is a direct outcome of society, which teaches men these stereotypes from an early age [80,81,82,83,84]. Boys and men are at risk of conforming to this desired image from unqualified agents or even from other boys or men who are unable to handle the masculinity figure due to all these cultural lessons being linked to aggression and violence. According to statistics, men are disproportionately represented in jails, and they also have a higher propensity than women to perpetrate violent crimes and women are most likely to become their victims. Depending on how they approach the situation, women’s roles in this conundrum can be a double-edged sword [80,81,82,83,84]. This tension of maintaining one’s masculinity lies at the heart of many aspects of AMSVo.

### 4.5. Incarceration and AMSVo

Both men and women experience pressured and coerced sex while incarcerated [85,86]. A common belief is that men are only raped in institutional settings such as prison [87]. Struckman-Johnson and associates (1996) found that 20% of those anonymously surveyed reported being forced to have at least one sexual contact while incarcerated. The range of sexual behaviors included intercourse (vaginal, anal, or oral) and sexual touching. Findings regarding the sexual violence of men during incarceration vary greatly. For example, Struckman-Johnson and Struckman-Johnson (2006) reported that higher rates of sexual assaults were found using anonymous surveys and lower rates in those studies that used interviews [66].

### 4.6. The Effect of Medical and Support Staff

Misconceptions are not only prevalent among the public, college students, and law enforcement but medical trainees, crisis workers, and counselors [17,50,52,65,66,88,89,90]. For instance, Kassing and Prieto (2003) studied counselors-in-training on the endorsement of rape misconceptions [52]. They found that counselor trainees accept some of the detrimental rape biases. They found that both male and female trainees expressed beliefs that people can do things to protect themselves from rape, thus implying a measure of responsibility for their victimization. One significant problem contributing to misconceptions about adult male victims is that it has remained virtually invisible (Kubiak 2018). The lack of accurate information and attention given to male victims by researchers, theorists, and media outlets has hidden the problem from view [91]. The first book addressing male rape was not written until the 1970s, and since then, only a handful of books and reviews have been published on male sexual victimization [92]. Despite these books on the topic [58,93], they have failed to capture widespread attention. Kassing and Prieto (2003) found that male counselor trainees were more likely to accept falsehoods surrounding rape to a greater extent than female trainees, especially if those male trainees had never counseled a sexual assault victim [52,65]. Another finding was that in terms of blame toward male victims, after accounting for trainee sex, age, social desirability, and experience with victims, participants perceived the nonresistant male rape victim negatively for not putting up more resistance against their attacker.

Studies that examined the level of blame assigned to men and women have shown mixed results. While some studies have found that male victims are blamed significantly more than female victims [94,95], results from other studies indicate women are assigned greater [96,97]. Other studies such as the findings of White and Robinson Kurpuris (2002) found that blame was attributed to a victim when participants held to more traditional gender-roles, particularly attitudes toward women and their roles in society and beliefs that men need status and respect [98].

The fact that the public associated AMSVo with homosexuality and prison populations obfuscates the reality that these crimes occur among heterosexuals, in the workplace, in dating relationships, and within other contexts. Knowledge and understanding are the best antidotes for ignorance.

## 5. Victim Responses

Whether male or female, victims are often not perceived with compassion and, in fact, may be blamed for their victimization [96]. Walker et al. found that when male victims do report, they often experience disbelief, hostility, and blame; such reactions hold true even when the rape is disclosed to friends and relatives. Furthermore, research seems to uphold the sex-role stereotypes associated with sexuality [56,57]. Research shows that both men and women respondents take male victims less seriously when the perpetrator is female than when the perpetrator is male [60]. Respondents typically judged such sexual assault as being mutual and associated with nominal stress and more enjoyment [96]. Even male respondents focus on the sexual act rather than the exploitative or assaultive nature of the act. In a study examining attitudes of male rape, Doherty and Anderson (2004 and 2008) engaged 30 male–female dyads in a discussion of vignettes of male rape [99,100]. They noted that men and women discussed the topic by differentiating the experiences of male and female victims and by stressing the similarity of the physical act of intercourse with rape. Accordingly, participants found heterosexual male rape as more disgusting than rape for homosexual men and women because it deviated from the normative heterosexual practice. This is consistent with Mitchell and colleagues (1999) who found that participants viewed homosexual male rape victims as more responsible, less traumatized, and finding the act more pleasurable [101].

The tendency of the public to blame victims is not unique to men but appears to exist at higher rates. Advocacy for men and greater public awareness are the best means of combatting the unseen victims.

## 6. Populations and Perpetrators of AMSVo

To date, most of the research on male rape has focused on institutional rape (e.g., correctional facility) and male-on-male rape. Studies on incarcerated men have found that their greatest fear is being raped in prison—a valid fear, because 7–12% of imprisoned men reported being raped an average of nine times [102]. Man and Cronan (2001) refer to the numbers of prisoner sexual assaults as “simply staggering” and “rampant” (p. 129). A selected number of studies have examined rape associated with homosexuals [22,103]. Examining the case of male-on-male rape, Vearnals and Campbell (2001) hypothesized that rape might be motivated by sex or by men wishing to demonstrate domination; interestingly, Man and Cronan (2001) hold that domination is the main motivation for rape in prison [104]. King and Woollett (1997) investigated men who sought counseling following their attack [105]. They found that 87% were assaulted by at least one man, 7% by women, and 6% by a man and a woman. Forced and anal penetration took place in almost all cases and 23% feared for their lives. In most of these cases, the assailant was known to the victim. Several studies investigated male rape by women perpetrators. Coxell and associates (1999) found that 46% of men reported rape or attempted rape by a woman [46]. Sorenson and colleagues (1987) found that white, college-aged men in a Los Angeles community census sample were most at-risk for experiencing sexual coercion and that most of the perpetrators were female acquaintances or lovers who used psychological pressure as opposed to physical force [106].

## 7. Risk Factors

The identification of variables that influence likelihood of AMSV is fundamental for prevention efforts (Loh et al., 2005) [107]. Ioannou, Hammond, and Machin (2016) classified sexual victimization into three general categories: the characteristics of the victim, the characteristics of the perpetrator, and situational characteristics (environmental, substance use, and violence factors) [108].

### 7.1. Characteristics of Victims

Studies have examined the characteristics of adult men who are sexually victimized. Ages ranged from 20 to 30 years old, were predominantly white, and considered vulnerable [35,109,110]. Vulnerability was noted in several studies, particularly in prison settings. According to Walker et al., a person’s routine and lifestyle influences the level of exposure one has to potential perpetrators and how vulnerable one is as a target. Studies are mixed on whether heterosexual men or homosexual men are more likely to be victimized [111,112,113,114,115]. The relationship between alcohol and victimization has been studied, leading to mixed findings. For example, Monks et al. (2010) found that victimization was related positively to alcohol risk scores and alcohol consumption-related problems. Men who drank were more likely to be victimized than those who did not. In contrast, Light and Monk-Turner (2009) found that 88% of men said that there was no substance abuse at the time of the assault [53].

### 7.2. Characteristics of the Perpetrators

Studies on the characteristics of perpetrators are lacking [108]. The importance of knowing these characteristics was noted by Loh and associates (2005) [107], who contend that the most effective way to reduce victimization is to focus on perpetrator characteristics. Dominance and power tendencies have been noted [109]. While some studies have found that most perpetrators are heterosexual [32], others report homosexuals as the most likely offenders [116]. Most studies have found that the offender is known to the victim [111,116], whereas Groth and Burgess (1980) interviewed offenders and found that 75% attacked strangers [103].

### 7.3. Environmental, Substance Use, and Violence Factors

Research on location has found that homes, car parks, and public parks were used frequently [35]. The relationship between alcohol and victimization has been studied, leading to mixed findings. Monks and associates (2010) found that men and women experienced similar levels of sexual victimization and that victimization was related positively to alcohol risk scores and alcohol consumption-related problems. In contrast, Light and Monk-Turner (2009) found that 88% of men said that there was no substance abuse at the time of the assault [53]. Since sexual assault is primarily regarded to be about sex, the eroticized rage and violence are ignored. The preponderance of research indicates that AMSV is more likely to be violent than that directed toward women [23], although some studies suggest that violence is greater toward women [112]. Du Mont and others (2013) found that male victims are subjected to significant violence, which includes strangulation, restraining, confinement, gagging, dragging, slapping, hitting, kicking, and biting [27]. Additionally, they found that about 20% report being forced to drink alcohol or use drugs prior to the assault and that 20% reported being attacked by more than one assailant. Sexually, male victims often experience sodomy and/or are forced to perform fellatio as well as genital touching and cunnilingus [89,101]. Ernst, Green, Ferguson, and Weiss (2000) found that men who are anally penetrated experience at least one form of rectal injury. Walfield (2018) notes that: “…results from the National Intimate Partner and Sexual Violence Survey (NIPSVS) indicate approximately a quarter of men (23.6%) experience some form of sexual violence over the course of their lifetime, which includes rape, being made to penetrate an individual, sexual coercion, and/or unwanted sexual contact” (p. 3) [59]. Use of weapons has ranged from 5% to 48.8% [32,53,114]. Weapons are less common when the offender is known to the victim [117], and rapes that are performed against incarcerated victims are less likely to involve weapons [66].

According to qualitative research on abusive gay relationships, psychological abuse is frequently present alongside physical and sexual assault [21,118,119,120,121]. As has been seen in heterosexual partnerships, different types of partner abuse can also occur in gay relationships. Even though several forms of abuse were considered in certain studies on gay partner abuse, only one prior study established associations between the various forms of abuse. In addition, there is a moderate to strong relationships between the victimization reports of various forms of abuse in a small convenience sample of gay men [21,118,119,120,121]. The fact that the act-based assessments of partner abuse do not take the victim’s experience into account is one of its limitations. Depending on the power and intent of the aggressor, as well as the victim’s reaction, the same violent act might result in several outcomes. As a result, the assessments of the effects of abuse, such as harm and need for medical care, must be added to measurements of abuse [21,118,119,120,121].

Public perception is that substance use is a significant contributing factor of any type of sexual violence [53,122]; yet research finds a wide discrepancy in the role that it plays. More research is needed that differentiates types of substances. Additionally, research is still required to determine what factors increase the offender’s perception of victim vulnerability, differences between methods used, and factors related to types of violence.

## 8. The Emotional Impact on Male Victims

### 8.1. Mental Health Problems

In the search focusing on male sexual victimization, distinct differences between male and female survivors began to emerge. As a result of victimization, men may experience a profound emotional disruption. Psychologically, victimization is especially trauma-producing for men [42,116]. Walker and colleagues interviewed 40 male rape survivors, leading to a detailed and descriptive analysis of the impact of the assault on men [56,57]. During the incident, men reacted with freezing, fear, helplessness, and submission. About 27% reported that they fought back without success. In the aftermath of the assault, they identified both short-term and long-term effects. Studies have also identified higher rates of mood disturbances, anxiety, suicidal ideation and behavior, non-suicide self-injury, grief and loss reactions, drug abuse, somatic problems sleep difficulties, sexual difficulties, increased changes in self-perception, social dysfunction, stigma, shame, lower self-esteem, hostility, fantasies about revenge, and an increase in a sense of vulnerability [41,53,56,57,58,89,116,123,124,125]. Post-traumatic stress disorder and rape trauma syndrome have also been noted. Self-blaming also affects how people respond to the victim, being perceived as less well-adjusted and more responsible for the rape than those who do not [32,114,126,127].

### 8.2. Response to AMSV

Despite the trauma associated with being raped, Sarrel and Masters (1982) found that men can respond sexually under disturbed emotional states, including anger and terror. They noted that men still experience post-trauma reactions [24]. A significant reaction immediately following the sexual trauma, as Frazier (1994) notes is anger because it is considered more masculine to respond that way [128]. However, many male victims reacted with a “controlled” style of coping exemplified by subdued reaction characterized by a calm, composed, acceptance and minimization of the assault. It is speculated that such a reaction reflects the aspect of male socialization to be emotionally inexpressive to aversive situations. Furthermore, Rogers (1998) suggested that this type of coping strategy renders male victims prone to long-term psychological problems as it makes help-seeking less likely, and denial undermines men coming to terms with their rape.

### 8.3. Decline in Relationship Satisfaction and Couple Intimacy

Studies also note problems with sexuality, sexual orientation, and sexual functioning [51,71,72]. One of the major impacts is on the male victim’s thoughts regarding his sexuality and sexual orientation. Many men report feeling less masculine, and some assume that they must be “gay” for another male to sexually assault them. Sexual dysfunction is also common, similar to the problems associated with male-to-female rape [24,41,51,56,57,71,72,116]. Specifically, research has noted impotence, anxieties over sexual activity, and increased promiscuity with women as a means of validating their sexual identity [41,56,57,116,129]. Somatically, men report poorer physical health than men who have not been victimized [129]. Specifically, men report numerous somatic symptoms, including tension, headaches, nausea, vomiting, ulcers, colitis, constipation and abdominal pain, fecal incontinence, decreased appetite and weight loss, and sexually transmitted infections [89,116,123,129,130].

## 9. Help-Seeking

Seeking professional help is especially difficult when the emotional distress is associated with poorly understood and misconceived ideas related to male victims of rape. Furthermore, AMS victims have fewer resources and greater stigma with sexual assault than female sexual assault victims [18]. Finkelhor (2008) notes that male victims face a lack of formal support systems [131]. The uniqueness of male victims requires clinicians to attend to the distinct issues surrounding their assaults. Light and Monk-Turner (2009) found that only 29% of men presented for help; such men were more likely to have been physically injured and have experienced penetration [53]. Despite 89% of male sexual assault victims reporting physical injuries, these researchers found that only 58% of male victims who reported being penetrated sought medical help. Pesola and others (1999) found that among men who sought help at an emergency room, 94% did so within 36 h of the assault [132].

For those who do seek help, Iseley and Gehrenbeck-Shim (1997) reported that most male victims went to rape crisis centers [32]. Of the 54 centers surveyed, only 15% reported having met with male victims. Most programs are ill equipped to provide adequate resources and appropriate staff to assist men, since services are designed for female, not male, victims. According to Washington’s (1999) study, only five percent of centers are prepared to meet male victim needs. This might explain the findings of Donnelly and Kenyon (1996), who interviewed 30 sexual assault treatment provides [17,133]. They noted that many rape crisis centers either explicitly refused services to male victims, were highly insensitive to their needs, and considered the issue inconsequential. Yet, these researchers noted very negative attitudes toward male victims. They made statements such as “…we don’t do men”, “men can’t be raped”, and “most males that are fondled or sodomized are males that wanted to be sodomized”.

## 10. Implications for Counseling

Given the paucity of literature addressing the treatment of male victims, counselors need self-awareness, training, and knowledge about the similarities and differences of male and female victims and must learn about the scope of treatment modalities and options.

### 10.1. Training on AMSVo

Mental health providers need to be able to correctly assess and identify gender-specific sexual trauma symptoms. Untrained therapists and those who fail to carefully screen and assess for the possibility of sexual assault on male victims are not likely to uncover it. Many graduate counseling programs do not require a human sexuality course. Without being educated, clinicians are primarily influenced by their own personal biases and counseling experience regarding whether and how to assess, treat, and demonstrate sensitivity to male victims. Davies (2002) contends that not only is research needed to identify the specific needs of men to improve service quality, but that providers go “above and beyond treatment for victimization generally” (p. 210) [25].

### 10.2. Counselor Awareness

Counselors need to self-examine their own beliefs, biases, and feelings [25]. The scope and depth of a counselor’s self-awareness is often proportionate to the ability to understand themselves through the experience of counseling [134]. Therapist self-insight helps a counselor distinguish between what is happening inside the self and what is happening with the client. Unaware counselors are vulnerable to being swept away by emotional tidal waves [134].

### 10.3. Creating a Safe-Haven

Male survivors would greatly benefit from being recognized as individuals who had suffered a devastating and traumatic event. A welcoming posture is especially critical for male victims [134]. Thus, working with men necessitates unique sensitivity to creating a haven for them to process the incredible and ineffable pain. When stories of sexual victimization are disclosed, counselors need to respond with warmth, compassion, empathy, and acceptance. Furthermore, connecting the client to his afflicted and distressful emotions will necessitate a haven. Along with this, normalizing the traumatic aftermath is another means of generating safety.

### 10.4. Counselor Assessment

Barriers to successful assessment is understanding the many reasons men remain silent, including unawareness, denial, the stigma of being a male victim, shame, fear, guilt, and fear of the listener’s reaction. Additionally, male victims may feel complicit if others perceive a lack of resistance against the attacker; shame and confusion over their body’s response to the assault; spiritual issues, especially if the perpetrator is a spiritual authority figure; and physical injuries associated with the attack. Young and associates inferred that unprompted disclosure was unlikely, placing the responsibility with the clinician to identify the assault [135]. Male victims were less likely to be asked about an abuse or assault. Read and colleagues identified several barriers to counselors inquiring about abuse and assaults [136]. Some of the barriers included: fears of vicarious traumatization, fears of inducing false memories, the client being male, and the client being more than 60 years of age. Agar and others identified several rationales for why it is crucial that mental health providers solicit information about a history of abuse: it allows the ability to conceptualize a client’s presenting problems more precisely; it enables the understanding of a client’s hesitancy to form a therapeutic alliance and build rapport; it helps to accurately identify hesitation instead of labeling it as resistance; and finally, it allows the understood client to receive an accurate diagnosis and, thus, appropriate treatment.

### 10.5. Validation of Experience

One role that helpers can play is supporting victims in acknowledging what has happened. While this is not completely unique to male victims, the misconceptions and lack of understanding can obfuscate the realities of what occurred. Donne and colleagues (2018) note that organizations are far more accustomed to supporting women in this regard [21].

### 10.6. Treatment Modalities

As a result of the unique impact on male rape victims and the range of possible reactions, longer-term and more specialized help is likely to be required [116]. The lack of research on male survivors, however, limits mental health providers’ ability to customize a male-oriented approach. Since most of the research on therapeutic effectiveness is focused on female victims, research on and clinical experience with female survivors can inform treatments for men. Cortois (1997) contends that since a combination of individual and group therapy has proven efficacious for female victims, it is likely appropriate for men [137]. Trauma models seem to be an effective approach to treatment.

### 10.7. Individual Therapy

Given the dearth of research addressing treatment strategies for male survivors of sexual violence, the literature concerning female survivors and adult men who experienced childhood sexual trauma can inform the treatment process. Male sexual survivors would benefit from identifying, labeling, and feeling feelings; cathartic exercises; learning adaptive coping strategies; anger management training; identity work; self-esteem approaches; boundary setting; approaches used with self-injurious behavior; treatments for co-occurring disorders and issues, such as depression, anxiety, substance abuse, sexual compulsivity, etc.; addressing sexual functioning issues; forgiveness of self and other work; intimacy building; and the promotion of safety strategies [138,139,140,141,142].

### 10.8. Group Therapy

Though individual treatments provide an avenue for expressing and modulating emotions, group therapy provides a common ground to facilitate feeling understood. A significant impact on men is isolation. The conflicting nature of the traditional views of masculinity and the culture of therapy contribute to male victims’ exclusion from the survivor community [143]. Without the connection to a healing community, men are abandoned to live in their misconceptions, distortions, and false beliefs. Group therapy provides an avenue for men to have those internal experiences identified and normalized. It is a vehicle toward shame-reduction. Group counseling allows for catharsis, which holds long-lasting benefits for male sexual survivors [144]. Moreover, group therapy offers the opportunity to use role-play and psychodrama. Both experiential methods offer opportunities to empower clients to act assertively, an often-counter narrative to the actual assault. Male victims can also act out existing or latent feelings within a non-judgmental and supportive environment [145].

### 10.9. Support Groups

General social support is often lacking for the male survivors of sexual assault when compared to female victims [40]. The lack of support from parents, family, and friends who are informed of the sexual assault creates a greater challenge for men. Kaufman et al. (1980) suggested that controlled reactions lead to men being emotionally inexpressive [146]. Accordingly, social support is critical of assisting male survivors in having their experiences normalized in a support milieu. While forming specific groups for male victims might be difficult geographically, online communities offer a means of connection. Davies (2002) advocates for law enforcement and medical staff being trained in making referrals to support groups that know how to help male victims [25].

## 11. Limits and Future Directions

Both quantitative and qualitative studies on adult male sexual violence are relatively few. The literature has poorly distinguished between different types of sexual violence, using the terms interchangeably. The United States’ Federal laws define sexual assault, rape, molestation, and other such terms as unique, although related, forms, which can obfuscate the drawing of meaningful conclusions for each type of sexual violence. Romantic partner sexual abuse, for example, is not clearly defined, although date rape is identified as its own form of violence. Many studies employ different research methodologies, including the use of very different participants who are drawn from community, student, and clinical groups. Victims who are studied in clinics may report higher distress and greater impairment compared to other groups.

Recommendations to enhance culturally relevant adult male sexual violence studies, including within-group cultural variability and comparisons against other populations across factors such as prevalence, impact, and treatment. Investigations need to consider an intersectional approach to assessing adult male victimization, which includes perspectives of gender identity, sexual identity, race, ethnicity, socio-economic status, and religious identification and commitment. Investigating offenders of AMSVo and the effects of societal and cultural responses requires attention. How similar and different are preparators of AMSVo from men who sexually offend against women?

All sexual trauma is clearly a serious public health problem. This review has several implications for advocates. It may provide a platform for providing awareness, prevention, and intervention training for law enforcement, the medical system, mental health practitioners, and other groups who have a connection to adult male sexual victimization. Such advocacy may open doors for collaborative work with all public systems to minimize the likelihood of secondary victimization and long-term distress. Since the public tends to believe that adult male sexual violence is a homosexual problem, education is critical to improving reactions to disclosure, reducing stigma, and raising awareness of available services and resources for survivors. Clearly, the greatest need for research is treatment. Identifying interventions that are male-specific will help treatment to be targeted and evidence-based. The full helping spectrum from support groups to trauma-based inpatient treatment, including attrition, requires investigation.

Practically, studies need to identify and define desired outcomes from increased efforts to address AMSV. Such outcomes are likely multiple, complex, and empirically supported. Research needs to speak to how to approach, assess, and address adult men prior to, during, and after the evaluation. By acquiring a common foundation of knowledge and fostering collaborations, those in the field may increase access to support and resources, so that all adult male survivors who experience the emotional aftermath of sexual trauma may follow a path of recovery that is healing and empowering.

## 12. Conclusions

Examining AMSVo is important because it is an issue that has been largely ignored in the literature. Given the fact that the stigma associated with men and the reluctance of them to disclose, it is not surprising that it has been a topic largely neglected in the literature. Much of the literature is old, dating back to the 1970s, when seminal research was taking place. Since that time, the empirical investigation of AMSV has been limited over the course of nearly 50 years. Given the lack of research into adult male sexual victimization, we do not have any certainty regarding what is applicable to today’s victims and what is outdated. The extant literature clearly dispels the misconceptions that men cannot be victimized outside of prison walls. According to this review, men frequently have erections or ejaculate uncontrollably during a sexual assault; nonetheless, the victim’s response does not imply consent. This review also supports the notion that anal rape might cause men to experience involuntary erections or ejaculations. These physiological responses complicate the impacts on men. It is as if their bodies are giving consent while their psychological selves feel violated.

Furthermore, current studies document the extensive negative impact on men. Advocacy and treatment efforts need to continue to be developed and researched, refined, and implemented to address the unique needs of male victims. When male victims are viewed as authentic members of the sexual survivor community, treatment and resources tailored to them should increase. Finally, the message that “rape is rape no matter who the victim is” needs to be emphasized.

## Data Availability

Not applicable.

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
