# Peer review of "Male Victims of Sexual Assault: A Review of the Literature"

_behavsci, 2023, doi:10.3390/bs13040304_

Round 1

Reviewer 1 Report

The paper is on a very important and under researched topic. As such, this comprehensive review is very timely and welcome.

Unfortunately, the paper is not well-written. There are numerous typos, missed words, repeat statements, and grammatical mistakes throughout. The formatting of references and in-text citations has many problems.

Throughout the paper, especially the first half, the authors need to be clearer on which findings come from empirical studies. Furthermore, when empirical findings are cited, especially major ones, please specify sample sizes and the nature of samples (forensic? General public?).

The second half of the paper (starting on line 331) is much stronger than the second half. I do not have any significant issues with the second half of the paper. The first half is not well organized and has several problems.

The section on prevalence statistics should be placed earlier in the paper and needs to be updated. The data that authors discuss is outdated. Both official statistics and findings from empirical studies should be discussed.

The paper could also benefit from a brief discussion of reasons why sexual assault of males has been understudied and of barriers to studying this topic.

Both The Perpetrators section and the Risk factors section are underdeveloped. Importantly, the paper ignores the fact that sexual assault could be perpetrated by romantic partners. I am suggesting some studies on the topic that the authors might consider incorporating (see further down).

The section on definitions is not sophisticated. First, why present a definition that is terribly outdated? (pp 57-58). Second, while it is acceptable to treat sexual assault and rape interchangeably for the review, the paper needs to explain to the reader the difference between these terms (and also “sexual abuse” and “sexual victimization”). How these terms are defined by the research community is different how they are defined by federal law enforcement.

The ”rape myths” section is a bit of a mess. First, it needs to be made clearer how research uncovering these myths has been done. What was the nature and size of samples? The general public? Law enforcement officers? Victims themselves? Second, the separation of the section into several subsections creates more confusion than clarity. For example, myths related to homosexuality and myths related to masculinity are very similar so it might make sense to put them into one sub-section. Discussion of myths related to incarceration is confusing. The narrative does not suggest that this is a myth so why call it a myth? Lines 157-172 do not really belong in the section where they are placed. Lines 207-214 list some of the same and some new myths. I strongly recommend rewriting/streamlining this section.

A number of very diverse consequences of sexual victimization are lumped together under “emotional impact.” (line 342). This section would benefit from sub-sections: increase in substance abuse problems, mental health problems, risky sexual behavior, decline in relationship satisfaction and couple intimacy have all been documented as consequences of sexual victimization (see suggested literature below).

Finally, it is not clear from the review, what is known about perpetrators of sexual violence on men. If little is known, perhaps, this could be identified as one of “directions for future research.”

Some suggested literature:

            Bartholomew, K., Regan, K. V., White, M. & Oram, D. (2008). “Patterns of abuse in male same-sex relationships. Violence and Victims 23(5), 617-636.

Hickson, C.I.H., Davies, P.M., Hunt, A. J., Weatherburn, P., McManus, T. J., & Cox, A.P.M. (1994). Gay men as victims of non-consensual sex. Archives of Sexual Behavior 23(3), 281-294.

Houston, E. & McKirnan, D.J. (2007). Intimate partner abuse among gay and bisexual men: Risk correlates and health outcomes. Journal of Urban Health 84(5), 681-690.

Messinger, A. M. (2014). Marking 35 years of research on same-sex intimate partner violence: Lessons and New Direction Pp. 65-84 in Handbook of LGBT Communities, Crime, and Justice. New York: Springer.

Stephenson, R., Rentsch, C., Salazar, L.F. & Sullivan, P.S. (2011). Dyadic characteristics and intimate partner violence among men who have sex with men. Western Journal of Emergency Medicine 12(3), 324-332.

Waterman, C. K, Dawson, L.J. & Bologna, M. (1989). Sexual coercion in gay male and lesbian relationships: predictors and implications for support services. The Journal of Sex Research 26(1), 118-124.

Author Response

Reviewer 1 Comments

The paper is on a very important and under researched topic. As such, this comprehensive review is very timely and welcome.

We appreciate the reviewer’s comment. We will expand on the edits given and improve upon the manuscript even further.

Unfortunately, the paper is not well-written. There are numerous typos, missed words, repeat statements, and grammatical mistakes throughout.

We appreciate the reviewer’s comments. We went through the manuscript and fixed any typos, missed words, repeat statements, or any grammar issues. These were shown in red.

The formatting of references and in-text citations has many problems.

-We appreciate the reviewer’s comment. We will use endnote and use another citation style to ensure the consistence of references in the manuscript.

Throughout the paper, especially the first half, the authors need to be clearer on which findings come from empirical studies.

-We appreciate the reviewer’s comment. We expanded upon the studies listed in the first half of the manuscript. We added information to the introduction, prevalence, definition, and barriers to reporting among sexual assault victims.

Furthermore, when empirical findings are cited, especially major ones, please specify sample sizes and the nature of samples (forensic? General public?).

-We appreciate the reviewer’s comments. We added specifiers to certain studies in the manuscript, particularly in the prevalence sections. We focused on two large studies of male sexual assault on college campuses (Ford and Edwards), which provided some of the larger samples for examining male sexual assault.

The second half of the paper (starting on line 331) is much stronger than the second half. I do not have any significant issues with the second half of the paper. The first half is not well organized and has several problems.

-We appreciate the reviewer’s comments. We made sure to organize and expand the content on the first half of the manuscript to help with the flow and connections to the later portions of the manuscript. We believe this information helped to improve the overall content and focus of the manuscript.

The section on prevalence statistics should be placed earlier in the paper and needs to be updated. The data that authors discuss is outdated. Both official statistics and findings from empirical studies should be discussed.

-We appreciate the reviewer’s comment. We added additional studies to this section from the international community and from recent studies on male sexual assault from college campuses.

The paper could also benefit from a brief discussion of reasons why sexual assault of males has been understudied and of barriers to studying this topic.

-We appreciate the reviewer’s comment. We discussed this very topic in the introduction and in a separate section to discuss the barriers to report sexual assault among male victims. 

Both The Perpetrators section and the Risk factors section are underdeveloped. Importantly, the paper ignores the fact that sexual assault could be perpetrated by romantic partners. I am suggesting some studies on the topic that the authors might consider incorporating (see further down).

-We appreciate the reviewer’s comments. We organized the section into the three subsections to improve reading clarity. In addition, we added references to the papers mentioned by the reviewer to the manuscript as well.

The section on definitions is not sophisticated. First, why present a definition that is terribly outdated? (pp 57-58). Second, while it is acceptable to treat sexual assault and rape interchangeably for the review, the paper needs to explain to the reader the difference between these terms (and also “sexual abuse” and “sexual victimization”). How these terms are defined by the research community is different how they are defined by federal law enforcement.

-We appreciate the reviewer’s comments. We added a large section discussing the general activities that fall under sexual assault. Then, we expand upon the technical definitions of sexual assault in the literature. Lastly, we add the legal definitions and changes that have increased awareness and acceptance of male sexual assault victims. We provided definitions of sexual abuse and sexual victimization to the introduction of the manuscript.

The ”rape myths” section is a bit of a mess. First, it needs to be made clearer how research uncovering these myths has been done. What was the nature and size of samples? The general public? Law enforcement officers? Victims themselves?

Second, the separation of the section into several subsections creates more confusion than clarity. For example, myths related to homosexuality and myths related to masculinity are very similar so it might make sense to put them into one sub-section. Discussion of myths related to incarceration is confusing. The narrative does not suggest that this is a myth so why call it a myth? Lines 157-172 do not really belong in the section where they are placed. Lines 207-214 list some of the same and some new myths. I strongly recommend rewriting/streamlining this section.

-We appreciate the reviewer’s comment. We made sure to define a rape myth as a misconception. We reorganized the section and added subdivisions to explore the major components of male sexual assault. We rewrote or removed the lines written above.

A number of very diverse consequences of sexual victimization are lumped together under “emotional impact.” (line 342). This section would benefit from sub-sections: increase in substance abuse problems, mental health problems, risky sexual behavior, decline in relationship satisfaction and couple intimacy have all been documented as consequences of sexual victimization (see suggested literature below).

-We appreciate their viewer’s comments. We broke the sections apart as suggested.

Finally, it is not clear from the review, what is known about perpetrators of sexual violence on men. If little is known, perhaps, this could be identified as one of “directions for future research.”

-We appreciate the reviewer’s comment. We added a section on directions for future research on male sexual assault.

Some suggested literature:

-We appreciate the reviewer’s comment. We added references to the manuscript.

            Bartholomew, K., Regan, K. V., White, M. & Oram, D. (2008). “Patterns of abuse in male same-sex relationships. Violence and Victims 23(5), 617-636.

Hickson, C.I.H., Davies, P.M., Hunt, A. J., Weatherburn, P., McManus, T. J., & Cox, A.P.M. (1994). Gay men as victims of non-consensual sex. Archives of Sexual Behavior 23(3), 281-294.

Houston, E. & McKirnan, D.J. (2007). Intimate partner abuse among gay and bisexual men: Risk correlates and health outcomes. Journal of Urban Health 84(5), 681-690.

Messinger, A. M. (2014). Marking 35 years of research on same-sex intimate partner violence: Lessons and New Direction Pp. 65-84 in Handbook of LGBT Communities, Crime, and Justice. New York: Springer.

Stephenson, R., Rentsch, C., Salazar, L.F. & Sullivan, P.S. (2011). Dyadic characteristics and intimate partner violence among men who have sex with men. Western Journal of Emergency Medicine 12(3), 324-332.

Waterman, C. K, Dawson, L.J. & Bologna, M. (1989). Sexual coercion in gay male and lesbian relationships: predictors and implications for support services. The Journal of Sex Research 26(1), 118-124.

Reviewer 2 Report

Congratulations! This article is a valuable review of the existing literature and commendable asset for professional workers. 

Author Response

We appreciate the reviewer's comment and taking time to go over the manuscript. 

Reviewer 3 Report

The paper deals with an interesting and overlooked topic.

Unfortunately, the structure as well as the methodology of the paper is more suitable for a book chapter than for a paper journal.

The authors should have used a more rigorous method, indicating keywords used for their research, focusing on a specific theme and using PRISMA guidelines for reviews.

Author Response

The paper deals with an interesting and overlooked topic. Unfortunately, the structure as well as the methodology of the paper is more suitable for a book chapter than for a paper journal. The authors should have used a more rigorous method, indicating keywords used for their research, focusing on a specific theme and using PRISMA guidelines for reviews.

-We appreciate the reviewer’s comments. We added a section at the beginning of the manuscript to define our search criteria we used for the manuscript to focus the specific theme and adhere to the general principles of the PRISMA guidelines. Our aim was not to do a systematic review. However, we wanted to be clear about how we came across the literature used in this manuscript.

Reviewer 4 Report

The article deals with a very interesting topic that needs to be explored in depth and gain a specific space for prominent analysis.

In order to be published, however, it needs to be reviewed in a substantive manner:

- The English needs to be revised there are typos and unclear periods (e.g. lines 123-126);

- It is unclear how the literature was handled: the part concerning the methodology for the construction of the literature to be examined is totally missing: how many articles were considered? on what basis were they selected? which databases were consulted? which keywords were used? which authors were the most significant? in which contexts was this subject considered?

- The definition of the objectives that guided the research is missing;

- Unfortunately, it is unclear why the selected articles considered the issue, so it is not clear whether the classification used was related to the objectives of the articles considered or an interpretation of the authors of the review;

- the use of the term 'mythology' highlights a lack of expertise in the field of social psychology. The terms 'bias' and 'stereotypes' are more correct. It is recommended that the authors also reconsider the literature on the basis of these keywords.

- It is advisable to construct a table illustrating the recognised articles and stereotypes:

- It is unclear whether the articles considered are affected by bias related to the stereotypes indicated as myths or have diagnosed them.

- The part concerning the listing of stereotypical representations and the part concerning applications, which, moreover, must be defined in the objectives of the survey, should be clearly separated.

- The part concerning the limits of the analysis made and the indication of future research is totally missing.

Author Response

Reviewer 4 Comments

The article deals with a very interesting topic that needs to be explored in depth and gain a specific space for prominent analysis.

In order to be published, however, it needs to be reviewed in a substantive manner:

-We appreciate the reviewer’s comment. We added additional content to the review and checked over the grammar and flow of the manuscript.

- The English needs to be revised there are typos and unclear periods (e.g. lines 123-126)

-We appreciate the reviewer’s comment. We added additional content to the review and checked over the grammar and flow of the manuscript.

- It is unclear how the literature was handled: the part concerning the methodology for the construction of the literature to be examined is totally missing: how many articles were considered? on what basis were they selected? which databases were consulted? which keywords were used? which authors were the most significant? in which contexts was this subject considered?

-We appreciate the reviewer’s comments. We added a section at the beginning of the manuscript to define our search criteria we used for the manuscript to focus the specific theme. Our aim was not to do a systematic review. However, we wanted to be clear about how we came across the literature used in this manuscript.

- The definition of the objectives that guided the research is missing

-We appreciate the reviewer’s comment. We wrote this paragraph in the introduction to set the objectives for the manuscript: “Based upon this process, this article provides a selective, but wide-ranging review of both narrative and empirical literature on adult male victims of sexual violence toward providing counselors knowledge about sexual violence against men and the unique challenges facing male victimso this end, the prevalence of male victimization, responses to it, populations that have been studied, types of perpetrators, risk factors, male reports of impact, and helping to seek will be addressed to further empower counselors to address challenges facing male victims.”

- Unfortunately, it is unclear why the selected articles considered the issue, so it is not clear whether the classification used was related to the objectives of the articles considered or an interpretation of the authors of the review

-We appreciate the reviewer’s comments. We added a section at the beginning of the manuscript to define our search criteria we used for the manuscript to focus the specific theme. Our aim was not to do a systematic review. However, we wanted to be clear about how we came across the literature used in this manuscript.

- the use of the term 'mythology' highlights a lack of expertise in the field of social psychology. The terms 'bias' and 'stereotypes' are more correct. It is recommended that the authors also reconsider the literature on the basis of these keywords.

- We appreciate the reviewer’s comment. We made sure to clarify the myth as a misconception/bias. The term rape myth is still used in recent literature. However, we can understand the reviewer’s concern on this. We made use to eliminate the use as much as possible form the manuscript.

- It is advisable to construct a table illustrating the recognized articles and stereotypes

-We appreciate the reviewer’s comments. We don’t believe a table would be necessary for this paper at the moment given the wide breadth of topics covered.

- It is unclear whether the articles considered are affected by bias related to the stereotypes indicated as myths or have diagnosed them.

-We appreciate the reviewer’s comment. We added additional information on the forms of bias that can be present in the studies on male sexual assault studies in the prevalence section and in the introduction.

- The part concerning the listing of stereotypical representations and the part concerning applications, which, moreover, must be defined in the objectives of the survey, should be clearly separated.

-We appreciate the reviewer’s comment. We added the objectives and focus at the end of the introduction in the following paragraph: “Despite the fact that adult male sexual assault is becoming more widely acknowledged as an issue, the literature unanimously agrees that there is a dearth of information on the subject when compared to information on female victims (Du Mont et al., 2013). According to this review, males frequently get an erection or ejaculate uncontrollably during a sexual assault; nonetheless, the victim's response does not imply consent. This review also supports the notion that anal rape might cause males to experience involuntary erections or ejaculations”.

- The part concerning the limits of the analysis made and the indication of future research is totally missing.

-We appreciate the reviewer’s comment. We will add a separate section before the conclusion to discuss limitations and future directions in the manuscript.

Round 2

Reviewer 3 Report

I think the present version of the paper is significantly improved. Congratulation to the author/s

I report just few little suggestions, especially I think the author could put a little deeper focus on elements of toxic masculinity, macho imagery and their consequences on men's health and life habits.

Line 29: a citation is missing in place of “CITE”

Line 30: a citation is missing in place of “CITE”

Line 84-85: physiology or psychology?

Line 195-196: repetition of the sentence

Paragraph 6: I suggest the authors expand a little the reference about stereotypical visions of men (masculinity) and women (femininity) according to the vision of “macho” imagery and toxic masculinity, as I think these are the roots of biases and misconceptions about AMSVo and many negative effects for men as well as for women. See for instance: doi: 10.1080/10532528.1991.10559871 ; doi: 10.1016/j.esxm.2020.03.003 ; https://doi.org/10.2979/jfemistudreli.33.1.23

Author Response

I think the present version of the paper is significantly improved. Congratulation to the author/s

-We appreciate the reviewer’s comment. Their comments helped improve the paper.

I report just few little suggestions, especially I think the author could put a little deeper focus on elements of toxic masculinity, macho imagery and their consequences on men's health and life habits.

Line 29: a citation is missing in place of “CITE”

We appreciate the reviewer’s comment. We used the citation (Basile et al., 2007).  

Line 30: a citation is missing in place of “CITE”

We appreciate the reviewer’s comment. We used the citation (Basile et al., 2007).  

Line 84-85: physiology or psychology?

-We appreciate the reviewer’s comment. Psychology; not physiology

Line 195-196: repetition of the sentence

-We appreciate the reviewer’s comment. We removed the repeat sentence.

Paragraph 6: I suggest the authors expand a little the reference about stereotypical visions of men (masculinity) and women (femininity) according to the vision of “macho” imagery and toxic masculinity, as I think these are the roots of biases and misconceptions about AMSVo and many negative effects for men as well as for women. See for instance: doi: 10.1080/10532528.1991.10559871 ; doi: 10.1016/j.esxm.2020.03.003 ; https://doi.org/10.2979/jfemistudreli.33.1.23

We appreciate the reviewer’s comments. We added a section expanding on the idea of macho imagery and toxic masculinity. We did this to help explain the fundamental basis of sexual assault among men. We also addressed some of the negative effects of toxic masculinity as well

Reviewer 4 Report

The article is now better written, more structured and better argued.

Author Response

The article is now better written, more structured and better argued.

Thank you for the response. We appreciate your feedback.